

# Cosmic censorship of trans-Planckian field ranges in gravitational collapse

Himanshu Chaudhary and Chethan Krishnan⋆

Center for High Energy Physics, Indian Institute of Science, Bangalore 560012, India

⋆ chethan.krishnan@gmail.com

## Abstract

A classical solution where the (scalar) field value moves by an $\mathcal{O}(1)$ range in Planck units is believed to signal the breakdown of Effective Field Theory (EFT). One heuristic argument for this is that such a field will have enough energy to be inside its own Schwarzschild radius, and will result in collapse. In this paper, we consider an inverse problem: what kind of field ranges arise during the gravitational collapse of a classical field? Despite the fact that collapse has been studied for almost a hundred years, most of the discussion is phrased in terms of fluid stress tensors, and not fields. An exception is the scalar collapse made famous by Choptuik. We re-consider Choptuik-like systems, but with the emphasis now on the evolution of the scalar. We give strong evidence that generic spherically symmetric collapse of a massless scalar field leads to super-Planckian field movement. But we also note that in every such supercritical collapse scenario, the large field range is hidden behind an apparent horizon. We also discuss how the familiar perfect fluid models for collapse like Oppenheimer-Snyder and Vaidya should be viewed in light of our results.

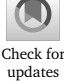

# 1  Introduction

Gravitational collapse is an old subject, and discussion on it is often phrased in terms of perfect fluid sources for the Einstein field equations[1]. This means that we describe the collapsing matter as being fully determined by the conservation law for the stress tensor, together with an equation of state (often dust or radiation, eg. [2]), and ignore the underlying (classical or quantum) field theory dynamics.

There exists one famous example where the full field dynamics at the classical level is incorporated into the study of gravitational collapse, and this is in the work of Choptuik and followers [3,4]. They studied the classical gravitational collapse of (an often massless) scalar field in numerical relativity. Their motivations for doing these were related to critical phenomena at the transition from dispersal to collapse, and will not concern us here. But their set up is one that we will also adopt. We will re-consider the gravitational collapse of a classical scalar field in Einstein gravity.

Our motivation for doing this arises from discussions of the swampland [5]. The swampland conjectures try to identify conditions that must hold in effective field theories, if they are to admit UV completion when coupled to gravity. One of these conjectures, called the Swampland Distance Conjecture (SDC) [6,7] implies that given an effective field theory action containing gravity and scalars, there is an $\mathcal{O}(1)$ range in the scalar field space for which the action is valid. Beyond that, a tower of light particles in the UV theory become light, rendering the effective field theory Lagrangian inapplicable. In other words, if a solution of the effective action predicts movement of the scalar beyond an $\mathcal{O}(1)$ range, then SDC claims that the solution is not reliable in those regions.

Evidence supporting the SDC in [6] (and many follow ups, eg., [8–11]), relied on string-inspired examples. But there also exists a heuristic horizon-based argument for the validity of SDC[2]. This goes as follows[3]: let a scalar field undergo movement by $\Delta\phi$ within a length scale of order $L$ in a $(d+1)$-dimensional spacetime. The energy contained in this region would go as $M \sim L^d \times (\Delta\phi/L)^2$ where we use the fact that energy density scales with the square of the field gradient and volume scales as $L^d$. From the Schwarzschild-Tangherlini metric, we expect that we will form a black hole when the energy $M$ within a region of size $L$ is such that $M/L^{d-2} \sim 1$. For the scalar field excursion above, this happens precisely when $\Delta\phi \sim 1$ in Planck units. We will call this the Heuristic Black Hole Argument (HBHA). See [8,18–20] for relevant discussions.

Our goal in this paper will be to reverse the logic of this and consider classical gravitational collapse sourced by fields. We want to know what kind of field ranges one can expect in gravitational collapse. This is a reversal of the previous reasoning, but note also one potentially significant conceptual distinction: in the way the HBHA is phrased, the $\mathcal{O}(1)$ field range happens essentially by construction. In the situation we consider, our starting point will be more conventional. It is the evolution of the system that will lead us to the trans-Planckian field range, when it happens.

As we noted previously, the context where this problem is well-posed is the one considered by Choptuik. Choptuik considered one-parameter families of scalar field profiles and identified

---

[1]See eg. [1], for a fairly conclusive paper from the late 30's.

[2]Swampland conjectures typically find their support in string theory arguments as well as in heuristic expectations about horizons. In fact, one can elevate this observation to a (partly tongue-in-cheek) Swampland Meta Conjecture (SMC): if one can find a claim which has evidence or inspiration in string theory constructions (see eg., [12]), and also has heuristic support in semi-classical gravitational settings typically involving horizons (see eg., [13]), then that claim is a swampland conjecture.

[3]This type of argument is connected to T. Banks, even though the precise origin of the idea seems to be lost in history. We thank a sci-post referee for some historical comments on this, and direct the reader to [14–17] for relevant discussions.

the critical parameter at which the configuration resulted in collapse as opposed to dispersal. Our goal will be different. We are not particularly concerned with the precise location of criticality in the parameter space, because we are not trying to figure out the scaling and self-similarity of near-critical configurations. We are primarily interested in the super-critical cases in Choptuik's set up, which result in the formation of true black holes and not just naked singularities. Also, our focus will be scalar field evolution in these systems unlike previous discussions that emphasized the metric. We will find that in near-critical and super-critical collapse scenarios the massless scalar field moves by an $\sim \mathcal{O}(1)$ field range. This result seems to be quite robust[4] as we vary the shape of the initial scalar profile.

Interestingly, we also find something more. We find that in every supercritical[5] situation where the field moves by $\mathcal{O}(1)$, the large field range is in fact covered up by an apparent horizon. Since apparent horizons are known to be inside or coincident with event horizons, this should be viewed as a type of cosmic censorship for the trans-Planckian field range in the EFT.

We will also briefly comment on well known perfect fluid collapse scenarios and how they tie up with our scalar field story. The key point as we will elaborate in a later section is that perfect fluid collapse solutions like Oppenheimer-Snyder or Vaidya, when realized in terms of scalar fields, contain field discontinuities. This need not imply that such stress tensors are inadmissible: our claim is merely that this means that they are not ideal for discussing super-Planckian field evolution. Since we wish to extract a clean statement about collapse and the field range bound, we will stick to the Einstein-scalar system.

## 2 Field Ranges in Scalar Collapse

For concreteness, we will consider the collapse of a scalar field minimally coupled to Einstein gravity [3]. This is a well-studied problem by now, but it is a numerical problem, so the conclusions are more accessible from the literature, than the details. We would like to have control on the details of our plots, so in what follows we evolve the system ourselves following eg., [21, 22, 25, 26]. The action takes the form

$$S = \frac{1}{8\pi G} \int d^4x \sqrt{-g} \left( \frac{1}{2}R - \frac{1}{2}\partial_\mu \phi \, \partial^\mu \phi \right), \tag{1}$$

where we have retained $G$ for ease of comparison with the conventions of other papers. The metric is of the form

$$ds^2 = e^{-2\sigma}(-dt^2 + dx^2) + r^2 d\Omega_2^2, \tag{2}$$

where $\sigma$ and $r$ are functions of $(t, x)$. The dynamical part of the equations of motion for this system can be taken in the form

$$r\Box r + (\nabla r)^2 = e^{-2\sigma}, \tag{3}$$

$$\Box \sigma - \frac{\Box r}{r} - \frac{1}{2}(\nabla \phi)^2 = 0, \tag{4}$$

$$\Box \phi + \frac{2}{r}(\nabla r . \nabla \phi) = 0, \tag{5}$$

---

[4]In fact because of this robustness, it seems inevitable to us that it must be implicit in previous work on scalar field collapse, even though its significance (as far as we are aware) does not seem to have been appreciated. Part of the problem is that since the work on Choptuik-collapse is numerical, one can only see from a paper what the authors emphasize. Nearby implicit results are not transparent.

[5]Note that the critical case has a naked singularity and violates cosmic censorship. Our observation applies unequivocally to super-critical cases with genuine horizons.

where the $\Box$ and the $\nabla$ are for the metric $-dt^2 + dx^2$. Note that we denote by $x$, the radial coordinate. Along with these, Einstein's equations also contain a set of constraints:

$$\partial_t \partial_x r + \partial_t r \, \partial_x \sigma + \partial_x r \, \partial_t \sigma + r \, \partial_t \phi \, \partial_x \phi \;=\; 0, \tag{6}$$

$$\partial_t \partial_t r + \partial_x \partial_x r + 2\partial_t r \, \partial_t \sigma + 2\partial_x r \, \partial_x \sigma + r\left((\partial_t \phi)^2 + (\partial_x \phi)^2\right) \;=\; 0. \tag{7}$$

Checking that the constraints hold will be a sanity check of our evolution. We should also choose our initial data to satisfy the constraints. But otherwise, they will not play a role in the evolution.

To evolve the system, it is useful to define a Meissner-Sharp mass $m$ via $e^{2\sigma}(\nabla r)^2 \equiv 1 - 2m/r$. This increases the number of variables in the PDE to four, but also increases the stability of integration [21, 22, 26]. The following are the PDEs that we will actually use in the code:

$$\partial_t m + \frac{r^2 e^{2\sigma}}{2}\left(\frac{1}{2}\partial_t r \left[(\partial_t \phi)^2 + (\partial_x \phi)^2\right] - \partial_x r \, \partial_t \phi \, \partial_x \phi\right) \;=\; 0, \tag{8}$$

$$\Box r - \frac{2m \, e^{-2\sigma}}{r^2} \;=\; 0, \tag{9}$$

$$\Box \sigma - \frac{2m \, e^{-2\sigma}}{r^3} + \frac{1}{2}(\nabla \phi)^2 \;=\; 0, \tag{10}$$

$$\Box \phi + \frac{2}{r}(\nabla r.\nabla \phi) \;=\; 0. \tag{11}$$

It can also be shown from the definition of $m$ that

$$\partial_x m - \frac{r^2 e^{2\sigma}}{2}\left(\frac{1}{2}\partial_x r \left[(\partial_t \phi)^2 + (\partial_x \phi)^2\right] - \partial_t r \, \partial_t \phi \, \partial_x \phi\right) = 0, \tag{12}$$

which will be useful in setting the initial data for our integrations.

## 2.1 Boundary & Initial Value Data

To evolve this hyperbolic system, we will need both boundary conditions at $x = 0$, and initial conditions at $t = 0$ for a fixed range which we will take to be $x \in [0, 2]$. We will not need boundary conditions at the other boundary of the domain ($x = 2$ here) because there is a causal boundary in spacetime defined by the ingoing lightsheet from $x = 2$. So the values on the other end of the domain can and will be set using extrapolation. If we were evolving the system for longer periods of time, or if our spatial domains were not large enough, this would not be possible. This is a standard practice [21, 22, 25, 26].

The boundary conditions at the origin are set by regularity. We first set the boundary conditions at $x = 0$, which is the center of the geometry. It is natural to set $r(t, 0) = 0$. This implies that $\partial_t r(t, 0) = 0$ and $\partial_t \partial_t r(t, 0) = 0$ as well. Now, in (5) we need the term $\frac{2}{r}(-\partial_t r \, \partial_t \phi + \partial_x r \, \partial_x \phi)$ to be regular at the origin. This forces us to set $\partial_x \phi(t, 0) = 0$ because $\partial_x r$ is not zero even for empty Minkowski space. Similarly, in (4) for regularity we require that $\frac{\partial_t^2 r - \partial_x^2 r}{r} = 0$ at $x = 0$, but because $\partial_t^2 r(t, 0) = 0$ we get $\partial_x \partial_x r(t, 0) = 0$. Finally, from (7) we get $\partial_x r \partial_x \sigma = 0$, and again using $\partial_x r \neq 0$ we have that $\partial_x \sigma(t, 0) = 0$.

To obtain the zeroth time step in our integration we will use the "corner" conditions $r(0, 0) = m(0, 0) = \phi(0, 0) = 0$ and $\partial_x r(0, 0) = 1$ and impose $\partial_t^2 r(0, x) = \partial_t r(0, x) = \partial_t \sigma(0, x) = \partial_t \phi(0, x) = 0$ at $t = 0$. These conditions are usually viewed as initial conditions that result in a moment-of-time-symmetry in the spacetime. With these "starting" conditions, we can compute/specify the initial data on the $t = 0$ slice for the variables $m(0, x), \sigma(0, x), r(0, x)$, and $\phi(0, x)$. We first specify the scalar initial profile to be some $\phi(0, x)$. This is essentially arbitrary, and in this paper, we will discuss four possible profiles for concreteness, whose parameters we will vary.

**Gaussian profile:**

$$\frac{\phi(t=0,x)}{\sqrt{8\pi}} = A \, \exp\left(\frac{-(x-x_0)^2}{\delta^2}\right). \tag{13}$$

**Maxwell (or Modified Gaussian) profile:**

$$\frac{\phi(t=0,x)}{\sqrt{8\pi}} = A \, x^2 \, \exp\left(\frac{-(x-x_0)^2}{\delta^2}\right). \tag{14}$$

**Ball (or tanh) profile:**

$$\frac{\phi(t=0,x)}{\sqrt{8\pi}} = A\left(\tanh\left(\frac{-(x-x_0)}{\delta^2}\right)+1\right). \tag{15}$$

**Shell profile:**

$$\frac{\phi(t=0,x)}{\sqrt{8\pi}} = A\left(\tanh\left(\frac{-(x-x_0)}{\delta^2}\right)+\tanh\left(\frac{(x-x_0+w)}{\delta^2}\right)\right). \tag{16}$$

Once the scalar initial profile is chosen, we also need to specify the initial data $m(0,x)$, $\sigma(0,x)$ and $r(0,x)$. These have to be chosen so that the Einstein constraints are satisfied. Using the "starting" conditions described above, we can use fourth order Runge-Kutta to evolve the equations (7), (9), (12) and get the zeroth time step (aka initial profiles). The values at the first time step can be obtained via a second order Taylor expansion using (5), (9), (10), and (8), or any single time step method like Lax-Wendroff method. (Note that we will need values at both zeroth and first time step, because we will be dealing with discretized second order PDEs.)

Once we have the zeroth and first time steps, the boundary conditions at $x = 0$ are enough to evolve the system, using a finite difference scheme to evolve the PDEs. The discretization scheme we use is as follows. In $X_j^n$ below, $X$ can be $\{\phi,\sigma,r,m\}$ and $n$ denotes a time step, while $j$ denotes a space step (step in $x$ direction):

$$X_{j,t}^n = \frac{X_j^{n+1}-X_j^{n-1}}{2\,\delta t}, \tag{17}$$

$$X_{j,x}^n = \frac{X_{j+1}^n-X_{j-1}^n}{2\,\delta x}, \tag{18}$$

$$X_{j,xx}^n = \frac{X_{j+1}^n-2X_j^n+X_{j-1}^n}{\delta x^2}, \tag{19}$$

$$X_{j,tt}^n = \frac{X_j^{n+1}-2X_j^n+X_j^{n-1}}{\delta t^2}. \tag{20}$$

By using the RHS of the above equations to replace the partial derivatives in our evolution PDEs, we can now solve them as a finite difference system.

## 2.2 Plots

Using the above set up, it is straightforward to evolve the Einstein-scalar system to see what kind of field ranges we see when there is collapse as opposed to dispersal[6]. The cleanest

---

[6]Note that capturing horizon formation is a somewhat tricky thing in numerical relativity. But because we are only after robust features and not delicate aspects (related to criticality), we can use the fact that the scalar disperses or not, as a measure of whether the black hole does (not) form. Any time the physics is near or above criticality, we expect to see large field movement.

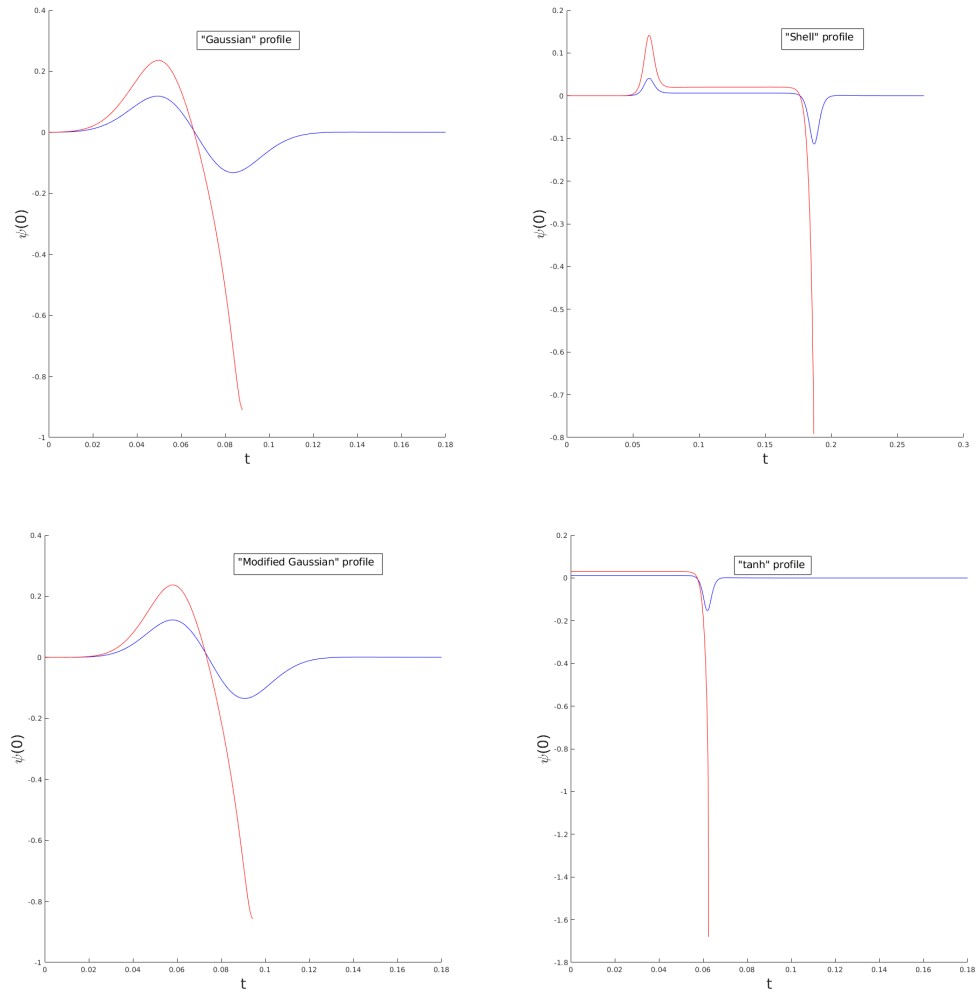

Figure 1: Plot of $\psi(0) \equiv \phi(t,0)/\sqrt{8\pi}$ vs. $t$ for some representative profiles. We have plotted a representative collapsing (ie., super-critical) case and a dispersing (ie., sub-critical) case of the same profile in red and blue respectively. The sub-critical cases start exhibiting large field movement as they get close to criticality, but otherwise their field movement is hierarchically smaller than $\mathcal{O}(1)$. All super-critical cases exhibit $\sim \mathcal{O}(1)$ field range. We have checked that the Einstein constraints are satisfied (within small numerical error) for every point shown in the curves.

demonstration of this can be found by plotting the scalar field value at the origin, as a function of time. We have stopped the plots when it is clear that the field has moved by[7] $\sim \mathcal{O}(1)$, but it is possible to see that this keeps increasing in many collapsing cases as we run the plots for a longer time. We suspect that the scalar goes on to take arbitrarily large values in magnitude, but this is of course difficult to show using numerical evolution. For illustrative purposes we consider a few profiles for the massless case in the figures, but less systematically we have also considered other profiles. In all the cases we considered, we have checked that the constraints stay small ($\sim 10^{-4}-10^{-3}$) for all points on the plots. For the Gaussian profile, even though we do not report it here, we have also checked the evolution with much higher precision. In the case of the ball and shell profiles, the fall is very sharp and we have stopped the evolution when the constraints reached above $\sim 10^{-3}$. This is enough to illustrate the $\sim \mathcal{O}(1)$ range. With

---

[7]Note that when the field changes sign, the $\Delta\phi$ takes that into account (as it should).

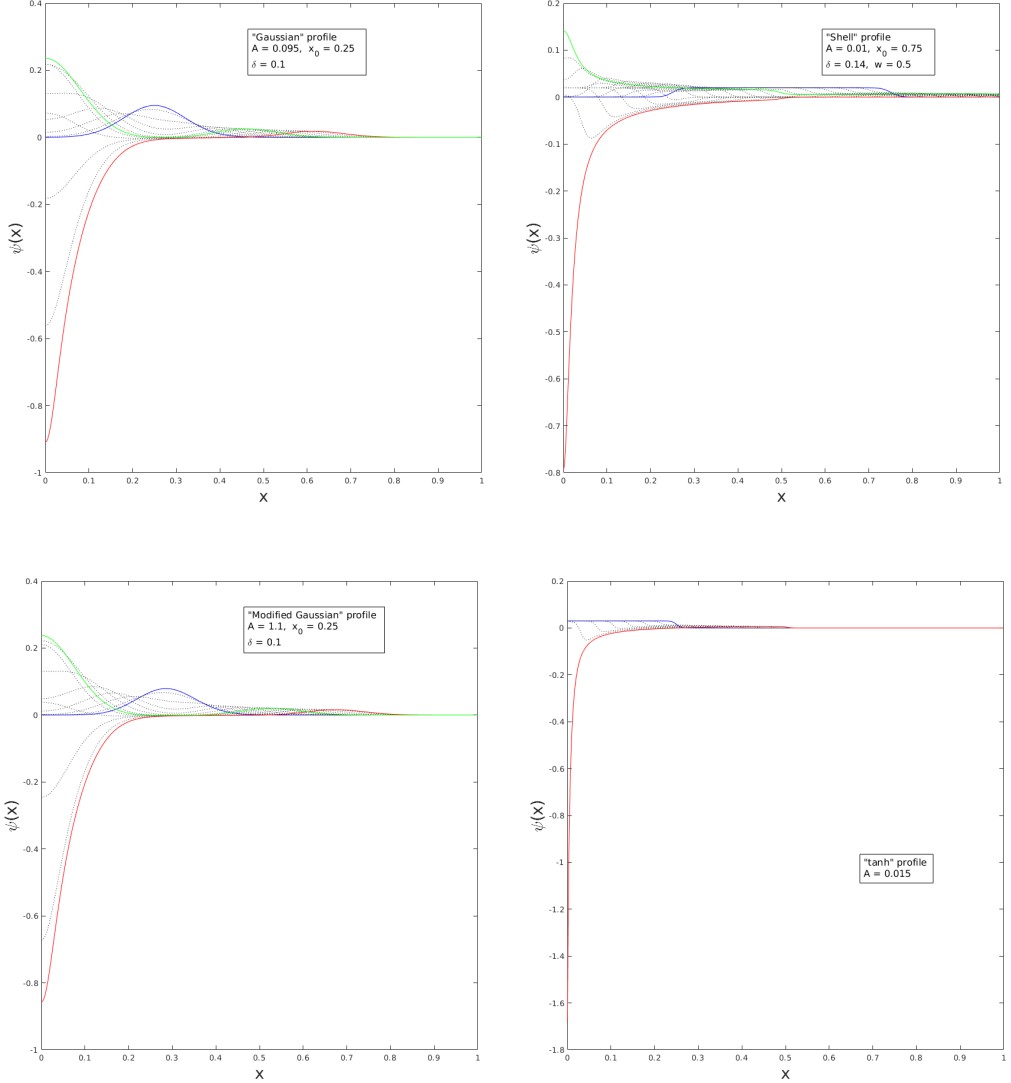

Figure 2: The spatial profiles at various times of different representative super-critical initial scalar profiles. We use the notation, $\psi(x) \equiv \phi(t,x)/\sqrt{8\pi}$. The initial profile is blue, the maximum value at the origin with the specific choice of parameters is attained for green, and red indicates a late-time profile.

more sophisticated numerical approaches (see eg. [23, 24]) and more powerful computers, it seems clear that one can go beyond this. For completeness, we also include the spatial profile plots of the scalar at various times in the above cases.

The explicit plots we present here are super-critical (ie., collapsing) cases, but fairly close to criticality. This corresponds to the onset of collapse and the $\mathcal{O}(1)$ field range, the sub-critical cases that disperse always have sub-Planckian field range and are not interesting for our purposes. We expect that the physics of super-Planckian field movement that we are capturing is directly related to collapse.

# 3 Apparent Horizon and Censorship of Trans-Planckian Fields

So far we have focused on the field ranges that arise during gravitational collapse, and established that generically when collapse happens the field moves by $\mathcal{O}(1)$. Now, we will discuss the location of the large field movement in relation to the apparent horizon. We see that in *all* the super-critical cases that we have looked at, precisely when the field starts moving wildly, an apparent horizon forms around it!

The relevant equation one has to solve in order to determine the location of the apparent horizon is (see eg. [25])

$$r - 2m = 0, \tag{21}$$

where $m$ is the Meissner-Sharp mass that we defined earlier. It turns out that at late times, the solution of this equation has two branches, see for example figure 5(a) of [25]. The branch with the bigger value of $x$ is the correct apparent horizon. We find that in all our super-critical cases, as long as we are not too close to criticality so that there is a clean separation of scales, the region where the field moves by $\mathcal{O}(1)$ is hidden behind the outer apparent horizon[8]. We present two representative plots here corresponding to a Gaussian profile with $\delta^2 = 2.5, x_0 = 10$ and $A = 0.1$. The figures are the spatial scalar profiles at two separate times. The first one shows the profile right after the onset of the large field movement. The apparent horizon has just appeared, and has barely split off into the two branches. The second profile is much later, and the two branches of the apparent horizon are clearly visible - note that the smaller branch is very close to the origin. The outer (true) apparent horizon covers up the large field range in both cases. This feature is generic within the various cases we have looked at.

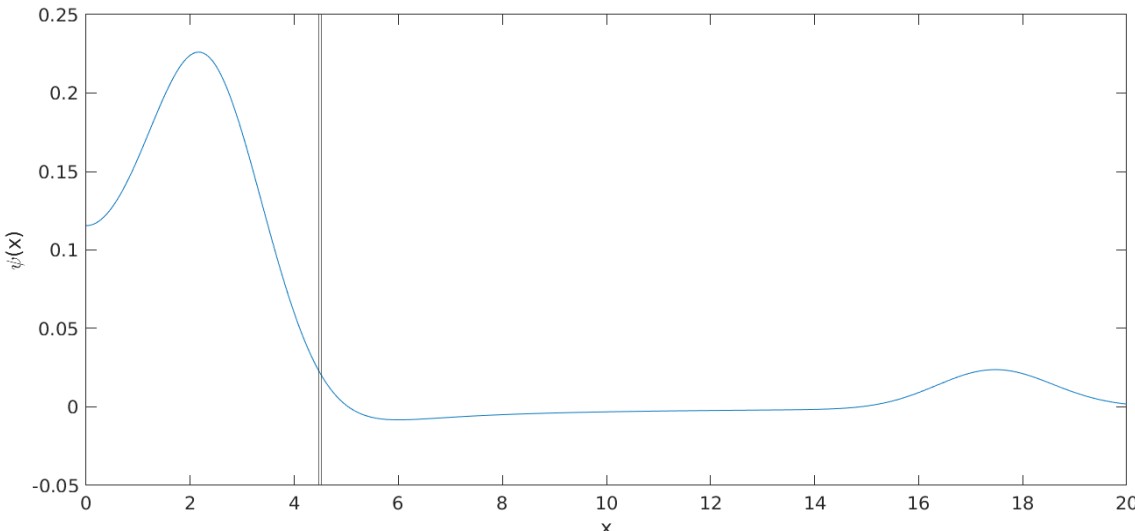

Figure 3: A scalar profile after the large field movement has just commenced. The vertical line(s) denote(s) the location of the apparent horizon.

---

[8]Note that it is crucial for an unambiguous conclusion that we are picking the correct branch of eqn. (21). In a previous version of this paper our conclusions were drastically different, because we did not pay the apparent horizon the respect it was due. We also thank a sci-post referee for asking a loosely related question.

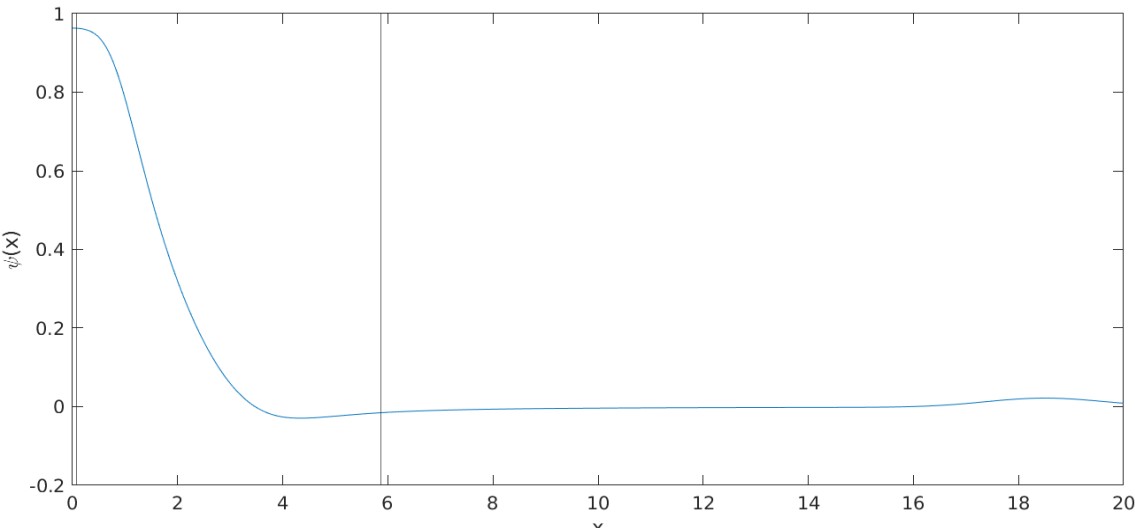

Figure 4: A late time scalar profile. Note the vertical line very close to the origin which is the spurious apparent horizon. Note also that the outer (true) apparent horizon covers up the whole region.

Let us make a comment about the critical case, before we close this section. When the collapse is very close to criticality, it is harder to separate the field movement from the horizon and the singularity, all of whose scales overlap with each other[9]. This is not surprising. Critical collapse is the boundary between collapse and dispersal, and corresponds to a mass-zero naked singularity that violates the letter (if not the spirit) of cosmic censorship. This is also a highly non-generic situation, because to reach it requires fine-tuning in the initial data. Since our observation above has the flavor of cosmic censorship, it is therefore unsurprising that we should see some inessential caveats when we ae close to criticality.

## 4 Perfect Fluid vs Scalar Field

A typical context in which one studies gravitational collapse in general relativity textbooks, is in terms of perfect fluid models. Since these are phrased in terms of stress tensors (whose dynamics is controlled by the covariant conservation law) and not in terms of the underlying fields, the claim about the field range bound cannot immediately be formulated there. But one can ask the question: what if we were to realize these perfect fluids in terms of scalar fields? How should we think of them in light of our discussions in the previous section?

We will argue that these fluid collapse models, when realized in terms of an underlying scalar field, lead to problems like field discontinuities. So in this paper we will not try to view them in terms of a coherent classical field.

Lets first consider Vaidya collapse, we will follow the discussion of p.644 in the lectures of [27]. One can try to realize the stress tensor (29.6) there in terms of the stress tensor of a (presumably massless[10], spherically symmetric) scalar field. Because the interior and the exterior are both vacuum solutions, a quick calculation reveals that this immediately leads to multiple difficulties. In particular, the vanishing of all but the $vv$ component of the stress tensor necessitates that $\phi$ should not be a function of $r$ and only be a function of $v$. But then

---

[9]Let us note here for completeness however, that even close to criticality, our numerical results indicate that most of field movement happens inside the apparent horizon.

[10]We have checked that allowing a potential for the scalar does not seem to improve the situation, because the vanishing of the $rr$ component of the stress tensor still forces the radial dependence of the scalar to vanish.

we also need

$$(\partial_v \phi(v))^2 \sim \frac{1}{r^2} \delta(v - v_0). \tag{22}$$

Mathematically, it may perhaps be possible to interpret the "square root" of the Dirac delta function by viewing the latter as a member of an operator valued Banach algebra. But this is unlikely to be physically meaningful: a sequence of Gaussians that converges to the delta function, converges to zero when one considers the square roots of those Gaussians. So it is not clear how to make this meaningful in a distributional sense. Equally problematically, the right hand side has an $r$-dependence. For these reasons, we will abandon our efforts to interpret the matter in a Vaidya collapse in terms of a scalar field.

Next, we consider the other canonical example of fluid collapse: the Oppenheimer-Snyder solution [1]. Here, we have a homogeneous FRW dust sphere in the interior and the exterior is empty Schwarzschild. To model dust we need a potential for the scalar field. The potential can be eliminated by setting pressure to zero for dust, which leads to the density $\rho = (\partial_\tau \phi)^2$ where $\tau$ is the FRW time in the interior. The scale factor evolution in the interior is controlled by the Friedmann equation which, together with $\rho a^3 = $ const, can be used to express $d\tau$ in terms of $a$ and $da$. These facts enable us to solve for $\phi$ as a function of $a$ (or $\tau$). This is doable, but we will not do this explicitly because the details are of no use to us. The crucial point for us is merely the observation that in the exterior, since the solution is vacuum Schwarzschild, the scalar must necessarily be zero (or constant). This implies that the scalar field necessarily has a discontinuity at the surface of the dust star leading to delta functions in its derivatives, which means that our original model of stress tensor is not satisfactory in the scalar field picture[11].

Let us make some further comments about some related points before we close this section. In Choptuik collapse of scalar coupled to gravity, the scalars have non-trivial radial dependence and evolution, unlike the step/delta function dependence of the fluid stress tensors we encountered above in analytic models of gravitational collapse. Indeed it is this non-trivial scalar profile that leads to the physics that we observed in the last section. So it will be interesting to try and model these fluid collapse models in terms of a suitable limit of a smoothed out scalar field to reproduce the results we observed previously. Let us also note that fluid stress tensors are supposed to be viewed in terms of a derivative expansion in the effective field theory, when the variations in the fluid are slow (the assumption of local equilibrium). This is again a suggestion that the fluid stress tensors with discontinuities that one often considers in collapse, need care in interpretation. The best way to interpret them is possibly in terms of a smoothed out discontinuity where the scale of smoothing is much larger than the UV cut-off (say mean free path in a weakly coupled fluid), but much smaller than the other length scales in the problem, say the size of the star[12].

## 5 Discussions

The results of this paper lead to two interesting observations. Firstly, when a scalar field undergoes collapse, very generically one encounters an $\mathcal{O}(1)$ field range, indicating the breakdown of effective field theory. Secondly, whenever that happens (and as long as we are away from the naked singularity corresponding to critical collapse) the large field range is hidden behind a horizon.

These observations immediately raise many questions. Let us address some of them below in lieu of a conclusion.

---

[11]Let us emphasize here that what we are talking about is not immediately about the singular shell stress tensors which arise from Israel junction conditions and such. This is about the discontinuity in the scalar.

[12]We thank Sayantani Bhattacharyya for explaining this to us.

- **Concern:** Perhaps the $\mathcal{O}(1)$ field range is the result of a coordinate choice?

  **Response:** Unlikely. Firstly, we are interested in the evolution of the scalar field itself, whatever the coordinates might be. A scalar field has a fixed value at any point in spacetime where we have laid down a chart. In some other coordinate system, that point will be labeled differently, but the value of the scalar field will remain unchanged. Therefore the statements about field ranges in the spacetime remain unaffected. Note also the tangential but noteworthy fact that we are considering a scenario where the horizon is supposed to form dynamically, but we are dealing with it numerically. So a Kruskal-like analytic extension will not be easy to construct. Let us also note that the system we are working with is a standard system in numerical relativity.

- **Concern:** Is the result a numerical artifact?

  **Response:** Unlikely. At every step in the evolution, we have checked the constraints. The evolution in the (thick) shell and the ball cases lead to pretty steep falls of the scalar (near collapse). But even there, the constraints stayed of $\mathcal{O}(10^{-3})$ – we stop the code/plot when it becomes bigger. In fact, in many cases we have checked the results to much higher accuracy. Note also that isolated examples of what we have reported about the field range is implicitly noted in pre-existing work. See for example figure 5 of [28] and (less obviously) figure 1.e of [22]. We expect that the result itself is unsurprising to the numerical relativity community, though a general statement of the type we are making seems to have not been made, nor has its consequences been recognized, because the focus has usually been on (near-)criticality. Note that what we are looking for is *not* a delicate phenomenon that is visible only at critical collapse. The claim is that collapse always leads to $\sim \mathcal{O}(1)$ field movement.

- **Concern:** How should one think about the well-known spherically symmetric fluid collapse scenarios? Like Oppenheimer-Snyder or Vaidya (see, eg., the detailed examples in [2, 27])?

  **Response:** This question has been addressed in some detail in section 3. The quick response is that our discussion is about classical (scalar) fields, and perfect fluids with discontinuities in the stress tensor are not immediately accessible in terms of smooth classical scalar fields.

  That said, it is not entirely clear to us what is the set up within which discontinuous fluid stress tensor configurations can be made sense of within EFT. We believe this question deserves more attention than what it has gotten until now. In particular, it will be very interesting to understand the underlying bulk field theory description of many of the shock wave constructions that exist in the literature, see eg., [29, 30]. One example[13] that comes tantalizingly close to a scalar field description of a shell/shock-wave is the set up in [31]. But it is perturbative, and not a fully backreacted system. Also the discussion is phrased in terms of scalar two point functions that determine the stress tensor expectation value in a QFT in curved space setting, so the language needs a bit of translation.

- **Response:** Kruskal co-ordinates exist for pre-existing black holes. Why does a classical scalar field propagating in a black hole background not necessarily see an $\mathcal{O}(1)$ field range, if one is working with Kruskal coordinates?

  **Response:** That deals with a probe scalar. Note that the problem we point out has to do with the formation of black holes due to scalar sources, which means that the scalar

---

[13]We thank Onkar Parrikar for reminding us of this work.

is highly backreacting. Note also that HBH Argument we presented in the Introduction cannot be made sense of in the probe limit.

Our result indicates that scalar field ranges may be a useful indicator of effective field theory breakdown, inside the horizon, at least in some contexts. This should be contrasted with the usual indicators of such a breakdown, namely the divergences in curvature scalars and tidal forces. Our work is in essence a converse of Banks' HBHA argument presented in the introduction.

It has previously been noted that cosmic censorship and Weak Gravity Conjecture are closely related, see eg. [32] and references in that work. Our observation seems to indicate that the connection may be broader. This raises the speculative possibility that perhaps classes of swampland violations of effective field theory can be hidden behind horizons. Other discussions of censorship (and its absence) in the context of SDC can be found in [18–20]. Note that our work deals with a dynamical collapse setting.

Let us conclude by mentioning some future directions that seem immediate and important. Firstly, we expect that working with non-canonical kinetic terms or more number of fields will also lead to a violation of the swampland distance bound, with the difference being that the field space distance should be suitably defined in terms of the appropriate field space metric. Note that in our case, the scalar value directly captures the distance in field space because we worked with a single scalar with a canonical kinetic term. More generally, to make the statement invariant under field space reparametrizations, one should work with the invariant field space distance.

A second point worthy of further understanding is the question of adding potentials to scalars. Preliminary investigations indicate that adding a potential/mass term does not change the claims we made about field ranges, it will be interesting to establish this more thoroughly.

Thirdly and perhaps most interestingly, even though the trans-Planckian movement of the scalar is censored by a horizon, we find that often the field movement happens before the origin. We have not tried to study this systematically, but it will be very interesting to see how the breakdown of EFT due to the field range ties with the other usual trackers of EFT breakdown (namely high curvatures, huge tidal forces, etc.). Note that the scalar range is a non-locally measured quantity in spacetime, whereas high curvatures are local. It will be remarkable if this fact can be put to some use in understanding the role of EFT in the black hole interior.

# Acknowledgments

We are grateful to Sayantani Bhattacharyya, Matthias Blau, Suvankar Datta, Jun-Qi Guo and Onkar Parrikar for discussions/correspondence. We thank Onkar Parrikar for comments on a version of this manuscript. We especially thank Jun-Qi Guo for sharing some of his code with us, and clarifications on it: this was instrumental in giving us confidence that our preliminary results were indeed correct.

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
