# Peer review of "Cosmic Censorship of Trans-Planckian Field Ranges in Gravitational Collapse"

_SciPost Physics, doi:SciPost Phys. 9, 036 (2020)_

## Round 1 · Referee Report · Anonymous (Referee 1) · 2020-7-8

Report

The authors study the swampland distance conjecture in the context of gravitational collapse. By numerically solving Einstein's equations coupled to a scalar field, they construct the back-reacted solution for various initial configurations of the scalar field and demonstrate that the scalar field deviates by O(1) in Planck units at late times. The authors suggest that this leads to tension with the expected smoothness at the horizon of the incipient black hole, as the distance conjecture would signal a breakdown of effective field theory. The paper is well written, explains all the technical details clearly, and the results are of broad interest to the high energy theory community. I would therefore recommend publication of the article.

One thing I found a bit unclear is the location of point in spacetime at which the scalar field becomes large relative to the horizon of the incipient black hole. For instance, if this point were deep behind the horizon, then there would be no tension with the validity of EFT at the horizon. So it may be helpful (at least for pedagogical pursposes) to show the location of this point on a Penrose diagram, or demonstrate clearly that this point is indeed causally connected with asymptotic infinity.

---

## Round 2 · Referee Report · Anonymous (Referee 1) · 2020-7-26

Report

In the new version of the manuscript submitted by the authors, they have argued that the gravitational solutions they found for scalar collapse always have an apparent horizon which hides the large field displacements behind it. This sufficiently addresses my question about the location of the region of large field displacement relative to the horizon of the incipient black hole, and also resolves the purported tension with the validity of EFT at the horizon in the previous version. As I stated in my previous report, the paper is clear, well-written, and the results are definitely of interest to the broad high energy theory community. I recommend publication of the paper without further changes.

---

## Round 2 · Referee Report · Anonymous (Referee 2) · 2020-8-9

Strengths

This is a very clear paper on an interesting and timely topic. It provides convincing numerical evidence that scalar field collapse probes super-Planckian field values, but that these values are hidden behind a horizon. It is well-written, informative and a pleasure to read.

Weaknesses

There are a few minor issues.

  1. The Swampland Distance Conjecture refers to the distance in scalar field space, not just the field value (which is not reparametrization invariant). It would be useful to emphasize that for the theory studied in this paper, the scalar field value is the same as the distance in field space, so the discussion really is phrased in terms of a physical, invariant quantity. For instance, the first bulleted concern in the Discussions asks about coordinate choices, but this concern applies to coordinates on field space as well as on spacetime.

  2. The discussion of the relationship to other work on super-Planckian spatial variations of fields could be improved. The authors mention a "Heuristic Black Hole Argument" and cite a 2008 paper of Nicolis, but they also say this is "usually attributed to T. Banks." I don't think that this attribution is quite correct. Banks discusses this at the beginning of section 3 of arXiv:1910.12817, where he says that "it was pointed out" to him at a Rutgers group meeting; unfortunately, he doesn't say who pointed it out. If there is something to cite before the paper of Nicolis, perhaps it is the related paper by Banks, hep-th/0011255, but it does not contain quite the same claims. Another, more tangentially related, early paper is arXiv:0705.2768 by Arkani-Hamed, Orgera, and Polchinski, which finds super-Planckian variations in some axionic wormhole solutions. (Although it predates the Nicolis paper, it mentions the basic "Heuristic Black Hole Argument" as if it was a well-known fact at the time.)

More recently, there have been a few papers that explore how large spatial variations of fields might be censored. One of these is cited as Ref. [8] (by Klaewer and Palti), but in the context of a list of references to the Swampland Distance Conjecture. I think it should be singled out as being closer to the topic of the current paper than the other SDC references. Similarly, arXiv:1701.05572 by Dolan, Draper, Kozaczuk, and Patel studies large spatial variations of axion fields around strings; arXiv:1901.00515 by Draper and Farkas studies large spatial variations of scalar fields in Kaluza-Klein bubble geometries; and arXiv:1910.04804 continues this study. The latter reference makes a conjecture about censorship of super-Planckian spatial field excursions, namely that they are bounded by $|\log(R \Lambda)|$, with $R$ the size of the spatial region over which the field varies and $\Lambda$ a UV cutoff on the theory. (This is consistent with examples of arbitrarily large field variations that are not screened by horizons, but arise only in tiny regions of space.)

I think that the introduction to this paper, and the paragraph in the conclusions that "raises the speculative possibility" that horizons hide large field variations, should refer to this existing body of literature. (In particular, there are known examples where the variations are not hidden behind horizons.)

  1. Reference [21] should supply an author and title, not just a web address.

Report

This paper should be published, after minor revisions to address the small issues noted under "Weaknesses." It is a high-quality paper that clearly meets the publication criteria of SciPost. It makes an interesting contribution to the field, in an area that many people are actively working on, and it does so in a concise, clear, convincing way.

Requested changes

See under "Weaknesses."

---

## Round 2 · Author Response

We thank the referee for the review. But unfortunately, despite his/her kind recommendation to accept with minor changes, we have made some substantive changes in the paper, including a change of title -- https://arxiv.org/abs/2003.05488

A key point is that at late times, there are two solutions to the apparent horizon equation and only the smaller one is inside the region of interest. Since the time of submission of the original version, we have noticed that in EVERY single case that is not too close to criticality, a bigger (and therefore correct) solution forms just outside, to cover up the large field range! This is a very strong suggestion that there is a cosmic censorship like mechanism in effect during gravitational collapse, that censors large field ranges.

While this result is quite interesting, it is a sharp deviation from our original punchline. While our original numerical results for the evolution are still correct, the interpretation changes completely.

We believe this also addresses the referee's query in the second paragraph - note that it is not very easy to construct Penrose diagrams of numerical gravity solutions, so we don't have a direct answer to his/her question. But the statements we have made above about the apparent horizon, together with the fact that the apparent horizons are expected to be inside global event horizons lead to a new observation -- trans-Planckian field ranges during collapse are always censored by a horizon.

So the main results of the paper are now -- (a) gravitational collapse of fields GENERICALLY lead to super-planckian field ranges, and (b) these field ranges are are always hidden behind apparent horizons.

The tension with the firewall is also gone.

---

## Round 2 · List of Changes

We have changed the title, a crucial sentence in the abstract, and removed/changed some paragraphs in the intro and conclusion to reflect our new perspective. We have also added a small section on the determination of apparent horizon, since it has turned out to be a more subtle and important issue than we expected.

---

## Round 3 · Author Response

We thank the referee for the report and the comments. We have made the following minor changes as suggested by the referee --

---

## Round 3 · List of Changes

1. In the earlier version, we had alluded to the case of multiple fields and non-canonical kinetic terms as a part of one of the final paragraphs of the paper. Now we have added a couple of more sentences, to emphasize the point referee makes. See the third last paragraph on p.14.

  2. (a) We have added the references that the referee pointed out regarding the "heuristic black hole argument" (HBHA), and edited footnote 3.

(b) We have mentioned [8] and the other three references pointed out by the referee, separately at the end of the paragraph that discussed the HBHA. The three references are now also mentioned as part of the conclusion, in the relevant paragraph.

  1. We have added the name of author and title to reference [27] (previously ref [21]).

---

## Editorial Decision

published